# Characteristics of the Colorless Polyimide-Based Flexible X-ray Detector with Non-Fullerene Acceptor Polymer

**DOI:** 10.3390/nano12060918

**Published:** 2022-03-10

**Authors:** Jehoon Lee, Jongkyu Won, Duhee Lee, Hailiang Liu, Jungwon Kang

**Affiliations:** 1Department of Electronics and Electrical Engineering, Dankook University, Yongin-si 16890, Korea; usyj0512@gmail.com (J.L.); 1jongkyu@naver.com (J.W.); clemenslee94@gmail.com (D.L.); liuhailiang107@gmail.com (H.L.); 2Convergence Semiconductor Research Center, Dankook University, Yongin-si 16890, Korea

**Keywords:** colorless polyimide, flexible detector, non-fullerene

## Abstract

In this paper, we investigate the characteristics of the colorless polyimide (CPI) film-based flexible organic X-ray detector. The CPI film can be applied to various applications, because it shows excellent visible light transmittance by removing the yellow color of polyimide (PI) film, which has the advantage of physical and chemical stability. In addition, the deformation curvature of the CPI substrate according to temperature showed similar characteristics to that of the glass substrate. For the organic active layer of the proposed detector, PBDB-T was fixed as a donor, and PC_71_BM and ITIC were used as acceptors. To evaluate the mechanical stability of the flexible detector, the degradation sensitivity was measured as bending curvature and bending cycle. The sensitivity of the detector with ITIC acceptor showed a 46.82% higher result than PC_71_BM acceptor on bending curvature (R = 10); and at the same curvature, when the bending cycle was 500 times, a 135.85% higher result than PC_71_BM acceptor.

## 1. Introduction

Recently, flexible devices are attracting attention from industry and consumers due to their lightweight, bendable, and non-breakable properties. When flexible plastic substrates are applied, solution process-based roll-to-roll and inkjet methods are possible, and economical advantages of facile manufacture can be obtained. Due to these advantages, they are applied in various applications, such as solar cells [1], organic light-emitting diodes (OLED) [2], organic thin-film transistors (OFET) [3], and radiation detectors [4]. In particular, radiation detector applications are found in non-destructive testing, security, and medicine. A commonly used radiation detector is the rigid flat plate-type, and when measuring curved human bodies and objects, image distortion occurs. Therefore, by applying flexible properties to radiation detectors, image distortion can be overcome in non-destructive testing, and in medical applications, such as dental, mammography, and computerized tomography. In addition, because of its flexible characteristics, it is convenient to store and carry, and has the advantage of being able to measure the image of an object at once, because it can be enlarged in a relatively easy process. Flexible radiation detectors can be manufactured using plastic substrates, such as polyethylene terephthalate (PET) [5], polyethylene naphthalate (PEN) [6], and colorless polyimide (CPI) [7].

X-ray detection methods can be classified into two detection methods: indirect and direct. The direct method has the advantage of high spatial resolution, but it is difficult to use inorganic materials in the process, such as silicon, amorphous selenium, and cadmium telluride. On the other hand, the indirect method has the advantage of high detection efficiency and since the active layer can be composed of organic or organic/inorganic semiconductors, it allows simple device fabrication, measurement, and evaluation [8,9]. Since the indirect method converts X-ray photons into visible light photons and absorbs them, the optical properties of the photon-charge conversion layer (called the active layer) are important. Inorganic materials mainly used in the active layer are expensive and difficult to form, thus research on X-ray detectors with organic materials whose optical properties are easy to change is in progress [10].

In this study, the active layer uses an organic semiconductor material that can be manufactured simply by solution processes, such as bar coating, spin coating, and inkjet printing. The organic layer is a bulk-heterojunction structure, in which a donor and an acceptor are mixed. Poly [[4,8-bis[5-(2-ethylhexyl)-2-thienyl]benzo[1,2-b:4,5-b′] dithiophene-2,6-diyl]-2,5-thiophenediyl[5,7-bis(2-ethylhexyl)-4,8-dioxo-4H,8H-benzo[1,2-c:4,5-c′]dithiophene-1,3-diyl]]) (PBDB-T), which has long chains and quinoids, has few charge traps and excellent carrier mobility [11,12], is used as a donor, while the commonly used (6,6)-phenyl C71 butyric acid methyl ester (PC_71_BM) is used as an acceptor, because it allows the formation of efficient bulk-heterojunctions to dissolve together with the donor polymer in common solvents [13]. However, the PC_71_BM acceptor has weak absorbance and is a fullerene derivative having a bucky-ball structure, thus there is a problem with poor mechanical stability [14,15]. Therefore, the fullerene acceptor is unsuitable for application to flexible detectors, so the replacement by a non-fullerene acceptor (NFA) should be considered in a structurally stretched and unfolded network structure. Among several non-fullerene acceptors, 3,9-bis(2-methylene-(3-(1,1-dicyanomethylene)-indanone))-5,5,11,11-tetrakis(4-hexylphenyl)-dithieno[2,3-d:2′,3′-d′]-s-indaceno[1,2-b:5,6-b′] dithiophene (ITIC) was used. ITIC has a push–pull structure in terms of chemical structure, and exhibits excellent properties, as well as carrier transport and light absorption [16]. As can be seen from the energy band diagram in Figure 1, the difference between the lowest unoccupied molecular orbital (LUMO) and highest unoccupied molecular orbital (HOMO) of ITIC and PBDB-T is within 0.4 eV, which reduces the exciton separation energy loss generated in the active layer compared to PC_71_BM, and could thereby improve the separated charge transferability [17]. The excitons generated from the donor by the visible photons were diffused and separated at the interface between the donor and the acceptor. After dissociation, electrons moved to the lithium fluoride/aluminum (LiF/Al) cathode and holes moved to the indium-tin-oxide (ITO) anode. To fabricate and evaluate a flexible organic radiation detector, the basic characteristics of glass, PEN, and CPI substrates were evaluated, the device was fabricated, and its performance was compared and evaluated through mechanical bending tests.

## 2. Experiment Set-Up

Figure 2a,b shows the organic X-ray detector fabrication process flow and detector image on glass and CPI substrate. The substrates on which the 2 × 2 mm^2^ active area was formed with a polyimide insulator on ITO patterned substrate was cleaned in an ultrasonic cleaner. For the glass and PEN substrate, acetone, methanol, and IPA were used, while the CPI substrate used only IPA. After being cleaned, it was baked in a vacuum oven for 10 min. UV-ozone treatment was performed to hydrophilize the surface of the ITO patterned substrate. Poly(3,4-ethylenedioxythiophene): poly(styrenesulfonate) (PEDOT:PSS) with dimethylformamide (DMF) as the hole transport layer (HTL) was spin-coated onto the ITO patterned substrate under 3000 rpm spin-rate condition. The thickness of the HTL was about 30 nm. The active layer solution was prepared by mixing PBDB-T and PC_71_BM (or ITIC) in a blend ratio of 1:1 using chlorobenzene solvent at a concentration of 20 mg/mL [18] and then spin-coated onto the HTL under 1100 rpm and 2500 rpm spin-rate conditions, respectively. The thickness of the PBDB-T:PC_71_BM and PBDB-T:ITIC was about 100 nm [19,20]. After that, it was baked on the hot plate for 10 min. LiF/Al (0.5/120 nm in thickness) as the cathode was deposited using thermal evaporation under a pressure of 10^−7^ torr. Finally, soda-lime glass was used as a cover material, and it was fixed with a UV-hardened sealant to prevent moisture and oxygen in the air.

Figure 3 shows the set-up for evaluating the flexible detector. After the bending test in a nitrogen atmosphere, soda-lime glass was attached to the detector for evaluation. A detector basically behaves similarly to a photodetector. Therefore, the evaluation method differs depending on the use or non-use of a scintillator that converts X-ray photons into visible-light photons. First, the scintillator-decoupled photodetector characteristics were evaluated using a solar simulator (San Ei Electronic, XES-40S2-CE). The detector was exposed to an AM 1.5 G filtered Xe lamp with 100 mW/cm^2^ illumination conditions, to extract generated current density–applied voltage (J–V) curves by applying a bias of −1.0 to +1.0 V. The detector without scintillator parameters for evaluation, such as short circuit density (J_SC_) and series resistance (R_S_) were calculated from the J–V curve. Then, the X-ray detector property of coupling the CsI(Tl) scintillator composed of 0.5 mm of aluminum and 0.4 mm of CsI(Tl) (rigid type Hamamatsu J13113 or flexible type Hamamatsu J1477) was measured using an X-ray generator (AJ-2000H, AJEX) and a source meter (Keithley 2400). The device and the CsI(Tl) scintillator were coupled using a jig. The X-ray operation condition was fixed with a voltage of 80 kVp, current of 63 mAs, and irradiation time of 1.57 s. The X-ray generator anode material was tungsten and the total filtration was 2.8 mm thickness aluminum equivalent. The voltage of – 0.6 V was applied to the detector anode (ITO) and cathode (LiF/Al) to collect the generated charges. The collected current density (CCD) was measured during the X-ray irradiation, while the dark current density (DCD) was measured without X-ray irradiation. The detection sensitivity, which is related to the photon-to-charge conversion efficiency, is an important factor of the X-ray detector, and was calculated using the following equation:CCD (μA/cm^2^) = (current during X-ray irradiation ON)/Detection Area(1)
DCD (μA/cm^2^) = (current during X-ray irradiation OFF)/Detection Area(2)
Sensitivity (mA/Gy∙cm^2^) = (CCD − DCD)/Absorbed Dose(3)

The mechanical flexibility and stability of the X-ray detector were measured using the bending machine. The degradation of detector performance was measured under different bending curvature (R) and bending cycle times. The bending curvature (R) was calculated according to Equation (4):R (mm) = (c^2^ + 4h^2^)/8h(4)
where, c is the curved distance of the bended detector and h is the curved height.

## 3. Results and Discussion

### 3.1. Experiments for Substrates Comparison

The properties of rigid glass substrates and plastic substrates, such as PEN and CPI, were compared. Figure 4a,b shows the deformation curvature and sheet conductivity of the substrate according to temperature. The glass transition temperature (T_g_) of the PEN substrate is about 160 °C, and when the heat of 150 °C is applied, the substrate begins to deform. Therefore, as the deformation curvature increases with increasing temperature, the sheet conductivity drops sharply. On the other hand, since the T_g_ of glass an CPI substrate is more than 300 °C, the temperature-dependent deformation curvature and deterioration of the surface conductivity properties are low. In general, since the firing temperature of the organic active layer is about 150 °C, it is important that the plastic substrate does not deform at a temperature below that. In addition, the transmittance of the substrate is also important for flexible detector applications. UV–Visible spectrophotometry (Optizen 2120UV) was used to show the transmittance according to the wavelength of each substrate, which is shown in Figure 4c. The transmittance of the CPI substrate in the visible light region is higher than that of the PEN substrate, and it is competitive, because it is similar to that of the glass substrate. According to the temperature-dependent deformation curvature, surface conductivity, and transmittance, if CPI substrate is used instead of a glass substrate, performance similar to that of a rigid organic material detector can be expected.

To evaluate the characteristics of each substrate, a device was manufactured using glass, PEN, and CPI, and the active layer was composed of PBDB-T as donor, and PC_71_BM as acceptor. Figure 5a shows the J–V curve of the photodetector without the CsI(Tl) scintillator, and parameters, such as J_SC_ and R_S_, were extracted from the detector J–V curve by J–V curve when an AM 1.5 G filtered Xe lamp with 100 mW/cm^2^ illumination conditions and voltages from −1 to +1 V were biased. The electrical characteristic, J_SC_, of the photodetector with PBDB-T:PC_71_BM (glass, PEN, CPI), was 12.68, 12.23, and 11.09 mA/cm^2^, respectively. Based on the glass substrate, the CPI substrate photodetector decreased by 3.55%, and the PEN substrate decreased by 12.54%. As an indicator of the uniformity of the active layer formation, R_S_ was 199.15, 204.20, and 322.17 Ω, respectively. The CPI was 2.54%, and PEN was increased by 61.77%, compared to the glass substrates. CCD, DCD, and sensitivity were measured by combining a rigid CsI(Tl) scintillator and a photodetector to which −0.6 V was biased under the X-ray operating conditions of 80 kVp tube voltage, 63 mAs tube current, and irradiation time of 1.57 s, as shown in Figure 5b. For the X-ray detector characteristics, the CCD of the CPI substrate decreased by 2.81%, and the sensitivity decreased by 4.07%, while the PEN substrate showed a decrease of 5.61% and 12.79%, respectively, based on the glass substrate. Compared to the glass and CPI substrates, the PEN substrates X-ray detector characteristics show relatively high degradation due to low J_SC_ and high R_S_. Therefore, the application of CPI substrate is competitive in the flexible detector, because it shows similar characteristics to the glass substrate.

### 3.2. Characteristics Experiment According to the Type of Acceptor Materials

According to the evaluation of devices manufactured using various substrates, the CPI substrate was fixed for flexible devices implementation, and experiments were conducted on different types of acceptors. The previously used PC_71_BM is a three-dimensional structure of a buckyball with fullerene derivatives, which has the disadvantages of high synthesis cost, low extinction coefficient, and low mechanical stability [21]. On the other hand, ITIC as a two-dimensional non-fullerene network structure has the advantages of mechanical stability, high carrier transport, and light absorption characteristics. Therefore, the PC_71_BM and ITIC acceptors were compared to find suitable materials for flexible detectors. The film condition was evaluated before application to the detector, and Figure 6a shows the AFM image. Through the AFM evaluation, the average surface roughness (R_q_) was 1.14 nm and 2.59 nm, respectively, depending on the different acceptors (PC_71_BM and ITIC), and Figure 6b shows other parameters, namely, R_S_ and J_SC_. As the R_q_ increases, the surface area for absorption increases, but the R_S_ including the surface resistance tends to increase, which as a result increases the R_S_ of the detector with the PBDB-T:ITIC active layer. The R_S_ was 204.2 Ω and 567.7 Ω according to the different acceptors. However, the J_SC_ of the detector with PBDB-T:ITIC was 16.02 mA/cm^2^, which shows a 31.10% improvement, compared to that of the detector with PBDB-T:PC_71_BM of 12.22 mA/cm^2^.

To verify the reason for the increase in R_q_ and R_S_ while the electrical properties improved, the electron-transport properties of the detectors according to different receptors were characterized, and the J–V curve of the detector with a bias voltage of –3 to +3 V under dark condition was shown in Figure 7a. Using curve fitting of the logarithmic J–V curve, the electron mobility (μ) and defect density were calculated using the space charge limited current (SCLC) model [22,23,24,25], using the Mott–Gurney Equation (5), and following the defect density Equation (6):Carrier Mobility (μ) = (8/9) × J × (L^3^/(V_a_^2^·ɛ_0_·ɛ_r_))(5)
N_defect_ (Defect Density) = (2·ɛ_0_·ɛ_r_∙V_FTL_)/(q·L^2^)(6)
where J is the current density and L is the thickness of the active layer (PBDB-T:PC_71_BM and ITIC) measured using a surface profiler (KLA-Tencor Alpha-step AS-500). The V_a_ is the applied voltage in the SCLC region, ɛ_0_ is the free-space permittivity constant of the vacuum (8.85 × 10^−^^12^ F/m), the relative permittivity (ɛ_r_) of the PBDB-T:PC_71_BM and PBDB-T:ITIC were 3.56 and 3.78, respectively [26]. The q is the elementary charge (1.602 × 10^−^^19^ C), and V_FTL_ is defined as the trap-filled limit voltage, which is the voltage at the point where the trap-limited SCLC region and the trap-filled limit region overlap on the log scale J–V curve.

The hysteresis according to the before and after X-ray irradiation of the fabricated detectors were measured, and it is shown in Appendix A. Figure 7b shows the mobility and defect density extracted from the Figure 7a graph measured in the dark condition to evaluate the charge-transfer characteristics of the fabricated detector. It also shows the CCD and sensitivity under X-ray irradiation condition, which is biased −0.6 V to the detector. The defect density is estimated to be 2.95 cm^−3^ and 2.08 cm^−3^ for the detector with PC_71_BM and ITIC acceptors, respectively. In addition, the electron mobility extracted from the detector with different acceptors was improved by 442.53% from 1.74 to 9.44 cm^2^/V·s for the PC_71_BM to ITIC acceptor, respectively. This is because the network structured ITIC lowers the recombination of charges in the active layer, and improves the electron mobility, due to the higher carrier transfer characteristics, compared to PC_71_BM. Therefore, when compared to the detector with PC_71_BM and ITIC acceptors, the surface roughness and R_S_ were increased, but the electrical property of J_SC_ was improved. After coupling the rigid CsI(Tl) scintillator with the photodetector, the CCD and sensitivity were evaluated. CCD was extracted from the X-ray detector with PC_71_BM and ITIC of 6.58 μA/cm^2^ and 6.95 μA/cm^2^ increase, respectively, due to the improvement of electrical properties, which is associated with the calculation of sensitivity, which showed a 14.55% increase from 1.65 to 1.89 mA/Gy∙cm^2^, respectively. Therefore, the surface roughness of the PBDB-T:ITIC active layer was non-uniform and the resistance higher than that of the PCBM acceptor, but it was confirmed that the detector performance was improved due to the improvement of the electrical properties, and the flexibility test was then evaluated.

A flexible scintillator was used to implement the flexible X-ray detector, as shown in Appendix A. When the flexible scintillator was applied, the sensitivity was 1.95 mA/Gy∙cm^2^. The sensitivity of the proposed flexible detector and several flexible detector studies were compared and shown in Appendix A.

### 3.3. Bending Curvature and Bending Cycle Experiments for Flexibility Evaluation

After evaluating the photodetector and X-ray detector according to the different acceptors, the characteristics were evaluated, such as the bending curvature (R) and bending cycles. First, Figure 8a,b shows the J–V curve of the photodetector with the PBDB-T:PC_71_BM and ITIC active layer according to the bending curvature test that was conducted. As the R increases, the characteristics of the device with PC_71_BM acceptor deteriorates sharply at R = 10, while the device with the ITIC acceptor shows a relatively low decrease. Table 1 shows the detector without scintillator parameters extracted from Figure 8a,b of J_SC_ and R_S_, respectively. J_SC_ values of the PCBM acceptor were 12.22 mA/cm^2^, 11.72 mA/cm^2^, 7.57 mA/cm^2^, and 6.83 mA/cm^2^ with degradation of up to 44.11% according to the bending curvature, while the R_S_ showed increase from 204.20 to 2604.55 Ω by 1175.49%. Meanwhile, the ITIC acceptor showed relatively low deterioration in J_SC_ by a decrease of 11.92% of R = 2, compared to the flat bending curvature, and an increase in R_S_ by 106.10%. As a result of the different acceptor characteristics, the 3D structure of PCBM has low stability of forming an active layer, which is brittle according to the physical bending, and the detector performance deteriorates rapidly, whereas ITIC has an unfolded structure, and is relatively more flexible than PC_71_BM, thus ITIC is suitable for the flexible detector.

The flexible X-ray detector was evaluated by combination with a flexible CsI(Tl) scintillator after the bending curvature experiment of the photodetector to measure the normalized sensitivity, as shown in Figure 9a. As the bending curvature decreased, the sensitivity of the ITIC detector decreased by 3.26%, 7.07%, and 6.52% from flat to R = 200, 10, and 2, respectively. On the other hand, the sensitivity of the PC_71_BM detector is reduced by 5.46%, 18.18%, and 12.12%, which is much higher compared to the ITIC acceptor. Among them, normalized sensitivity showed a 46.82% difference in comparison with R = 10 according to the different acceptors, which showed a significant decrease in performance, and a 61.83% difference at R = 2. Next, a bending cycle experiment was performed by fixing the bending curvature of R = 10, the performance of which was significantly reduced, and Figure 9b shows the results. As the bending cycle increases, the performance deteriorates due to the continuous mechanical stress, and when increasing the bending cycle from 100 to 500 times, the normalized sensitivity of the X-ray detector with the PPBDB-T:PC_71_BM active layer was 13.78% and the PBDB-T:ITIC active layer was 3.42%, showing relatively low degradation characteristics. Under the condition of bending curvature of 10 mm^−1^ and bending cycle of 500 times, the X-ray detector with PBDB-T:ITIC shows 135.85% higher performance than the PBDB-T:PC_71_BM active layer, thus a non-fullerene structured material, such as ITIC, is suitable as an acceptor for the flexible X-ray detector.

## 4. Conclusions

In this study, we investigated the characteristics of the colorless polyimide (CPI) film-based flexible X-ray detector. Plastic substrates, such as polyethylene terephthalate (PET), polyethylene naphthalate (PEN), and polyimide (PI), were mainly used for the flexible devices. The CPI film has an advantage in sensor applications, because it shows excellent transmittance by removing the yellow color from PI, which has physical and chemical stability. In addition, compared with the PEN substrate with high thermal stability, the deformation curvature according to the temperature showed similar characteristics to that of the glass substrate. Therefore, CPI was applied to the proposed flexible X-ray detector, due to having flexibility and similar characteristics to the glass substrate. The active layer of the detector was composed of an organic donor, such as poly[(2,6-(4,8-bis(5-(2-ethylhexyl)thiophen-2-yl)-benzo[1,2-b:4,5-b′]dithiophene))-alt-(5,5-(1′,3′-di-2-thienyl-5′,7′-bis(2-ethylhexyl)benzo[1′,2′-c:4′,5′-c′]dithiophene-4,8-dione)] (PBDB-T), and an acceptor, such as (6,6)-phenyl C71 butyric acid methyl ester (PC_71_BM), as a bulk-heterojunction (BHJ). However, PC_71_BM is a three-dimensional (3D) bucky-ball structure that is easily broken by physical bending, so that is not suitable for flexible devices and has the disadvantage of low morphological stability in mixed organic materials. Therefore, the PC_71_BM acceptor was replaced with a non-fullerene acceptor, such as ITIC, with a two-dimensionally unfolded structure, and has high carrier transport and light absorption characteristics. The detection sensitivity of the detector with PBDB-T:ITIC was higher than that of PBDB-T:PC_71_BM by 11.18%. The mechanical stability evaluation of the flexible detector measured the decrease in detection sensitivity through the bending curvature (R) and the bending cycle. First, the normalized sensitivity of the detector using the PBDB-T:ITIC active layer was 46.82% better than that using the PBDB-TT:PC_71_BM active layer at the bending curvature of 10 mm^−1^, by fixing the 50 times bending cycle. Second, the normalized sensitivity change was measured according to the bending cycle 500 times with the bending curvature fixed at R = 10, where the detector using the PBDB-T:ITIC active layer decreased by 3.42%, and that using the PBDB-T:PC_71_BM active layer decreased by 13.78%; and the normalization sensitivity of the PBDB-T:ITIC active layer gives better results by 135.85%. Therefore, the non-fullerene acceptor ITIC is more suitable for a flexible device, because it has better mechanical flexibility than PC_71_BM.

## Figures and Tables

**Figure 1 nanomaterials-12-00918-f001:**
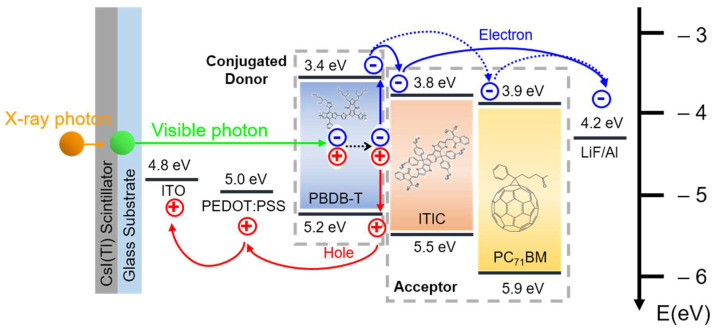
Energy band-diagram of the proposed organic X-ray detector with different acceptors as PC_71_BM and ITIC.

**Figure 2 nanomaterials-12-00918-f002:**
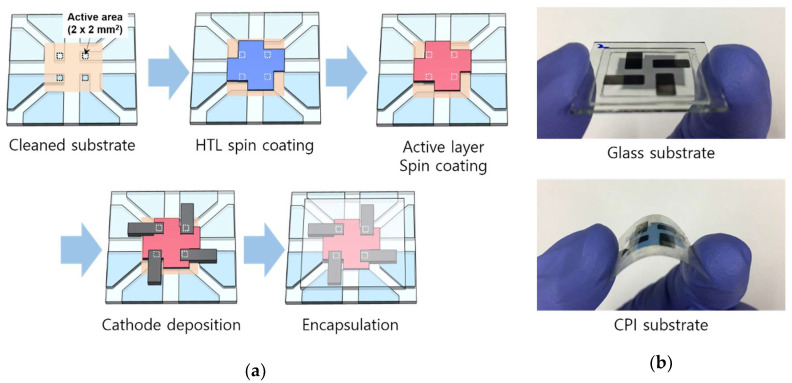
(**a**) Fabrication process of organic detector, and (**b**) images of fabricated detector on glass and CPI substrate.

**Figure 3 nanomaterials-12-00918-f003:**
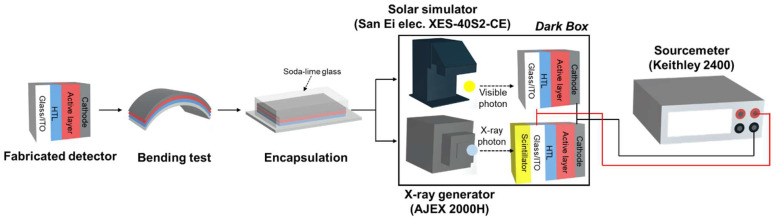
Experimental set-up of measuring the photodetector parameters of the scintillator decoupled detector, and the X-ray parameters of the scintillator coupled photodetector.

**Figure 4 nanomaterials-12-00918-f004:**
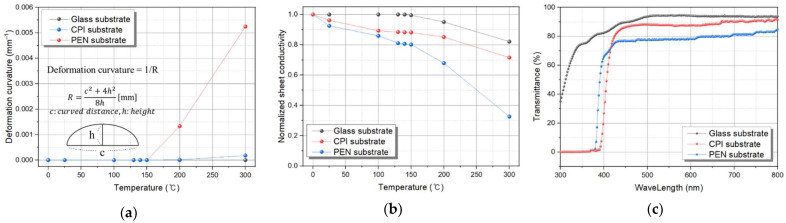
(**a**) Deformation curvature of glass, PEN, and CPI plastic substrate, (**b**) sheet conductivity of the substrates according to heat temperature, and (**c**) transmittance of substrates.

**Figure 5 nanomaterials-12-00918-f005:**
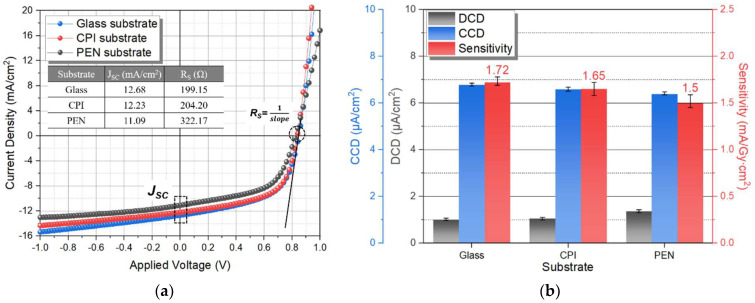
(**a**) J–V characteristics of photodetector with the PBDB-T:PC_71_BM active layer, and (**b**) CCD, DCD, and sensitivity of the X-ray detector on glass, CPI, and PEN substrates.

**Figure 6 nanomaterials-12-00918-f006:**
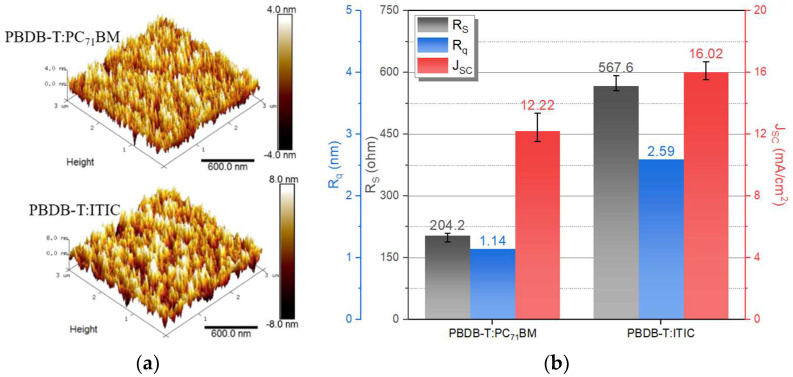
(**a**) AFM images of the organic active layer films, (**b**) average surface roughness (R_q_), R_S_, and J_SC_ with different acceptors (PC_71_BM and ITIC).

**Figure 7 nanomaterials-12-00918-f007:**
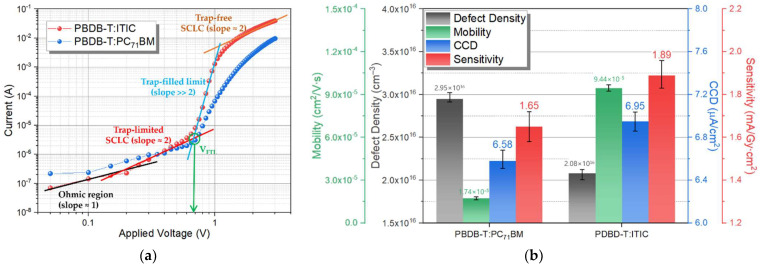
(**a**) Curve fitting of the logarithmic J–V characteristics, and (**b**) the mobilities, defect densities, CCD, and sensitivity of the as-prepared detector with the different acceptors under the dark condition.

**Figure 8 nanomaterials-12-00918-f008:**
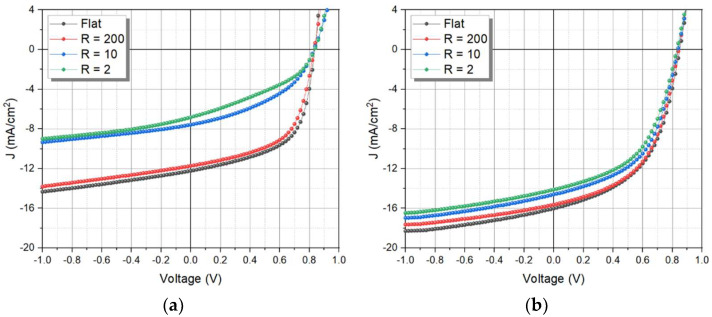
J–V characteristic of the detector according to bending curvature (**a**) with PBDB-T:PC_71_BM, and (**b**) PBDB-T:ITIC, fixed at 50 times bending cycle.

**Figure 9 nanomaterials-12-00918-f009:**
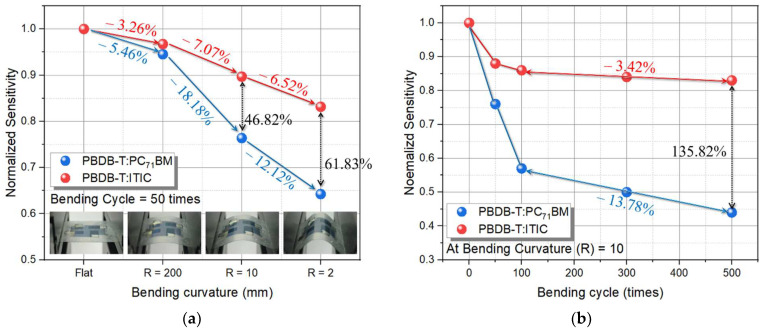
Normalized sensitivity degradation of the X-ray detectors as a function of (**a**) bending curvature, and (**b**) bending cycle times.

**Table 1 nanomaterials-12-00918-t001:** The parameters of detector without CsI(Tl) scintillator according to different acceptors (PC_71_BM and ITIC).

Active Layer	Bending Curvature	J_SC_ (mA/cm^2^)	R_S_ (ohm)
PBDB-T:PC_71_BM	Flat	12.22	204.20
	R = 200	11.72	479.43
	R = 10	7.57	1925.18
	R = 2	6.83	2604.55
PBDB-T:ITIC	Flat	16.02	567.60
	R = 200	15.64	617.41
	R = 10	14.61	851.14
	R = 2	14.11	1169.84

## Data Availability

The data is included in the main text and the Appendix A.

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
