# Peer review of "Characteristics of the Colorless Polyimide-Based Flexible X-ray Detector with Non-Fullerene Acceptor Polymer"

_nanomaterials, 2022, doi:10.3390/nano12060918_

Round 1

Reviewer 1 Report

In this paper, flexible X-ray detectors have been fabricated. The colorless polyimide film shows better performance of the most common PET flexible substrate. To evaluate the mechanical stability of the flexible detector, the deformation curvature of glass, PEN and CPI plastic substrate have been evaluated. Moreover the sheet conductivity of the substrates according to heat temperature, and transmittance of substrates show a better behaviour of the CPI substrates, more similar to the glass. About the acceptor optimization, it results that the sensitivity of the detector with ITIC acceptor showed significant higher result than PC71BM acceptor on bending curvature. The conclusions are coherent with the results, for this reason the manuscript could be accepted with minor revisions:

  • Explicit the acronyms in the abstract and in the rest of the manuscript the first time you mention them (like CPI, ITIC and so on…);
  • Experimental section: could the author describe in detail the layer deposition and the encapsulation method used? Is the area of the HTL and delimited by a mask?
  • Are this detectors stable and without histeresys? J−V characteristics acquired in dark conditions, before and after the X-ray exposure could be added.
  • All the data (JV, sensitivity…) should be reported as statistics, how many samples have been measured? Could the author add the statistic?

Reviewer 2 Report

The paper describes the fabrication and characterization of the flexible organic X-ray detector based on colorless polyimide (CPI) with the addition of different donor and acceptors. An improved (over conventionally used PEN substrates) detector’s mechanical stability after bending and photoconductivity was demonstrated, and a suitable acceptor for the flexible detector was proposed. The performance of the proposed X-ray detector was shown in conjunction with a CsI scintillator.

The article is well-written, results and conclusions are clearly stated.

Some specific comments:

Abstract: The authors should write the abbreviations in full (CPI, PI).

Lines 69-73: A relevant reference to the energy levels would be appropriate here.

Section 2. Experiment Set-up: What is the thickness of each layer? How thick is the scintillator? Some details about the X-ray tube would be appropriate (anode material, filtration). What was the X-ray exposure to the detector?

Line 83: What kind of glass was used (soda lime, fused silica, etc.)?

Line 104: Authors should explain how the scintillator was coupled to the substrate in more detail.

Line 108: It is unclear which electrode (ITO or LiF/Al) the voltage was applied.

Line 109: “the dark current density (DCD) was measured during X-ray non-irradiation” perhaps will sound better as “the dark current density (DCD) was measured without X-ray irradiation”.

Equation 3 and through the text and figures: “Sensitivity [mA/Gy∙cm2]” – it should be clearly indicated that cm2 is in the denominator. Also, the X-ray sensitivity is usually characterized as a charge (Coulombs) per unit area per unit dose, not as a current (Amps). The authors should use a conventional term or explain why Amps would be preferential over Coulombs in this work.

Line 116: “according to the following Eq. (2)” should be Eq. (4).

Line 127: capitalize CPI.

Lines 145-146, 170: the abbreviations for PBDB-T, PC71BM and ITIC have already been explained before; perhaps there is no need to repeat them.

Lines 166-168: A relevant reference would be appropriate here.

Line 195: what is the value of permittivity er of the active layer?

Figure 7(a): A consistent numbering style on the left axis is recommended.

Line 251: “As the bending curvature …” a word is missing (increases?).

Reviewer 3 Report

The paper presents an indirect flexible X-ray detector based on organic substrate and an active layer. It is a topic of interest to researchers in the photoelectric areas. However, the main problem of this article is that the logic is confused, and some of the data presented seem inadequate to support the argument. The paper needs very significant improvement before acceptance for publication. Detailed comments are as follows:
1. As the energy band diagram in figure 1, excitons are produced at the junction of donor and acceptor. The author should confirm the location of exciton generation and dissociation in the heterojunction.
2. On page 3 line 96, “The thickness of the PBDB-T:PC71BM was about 100 nm and PBDB-T:ITIC was about 120 nm.”, the functional layer of the device based on PBDB-T:ITIC is 20% thicker than that of PBDB-T:PC71BM, but the detection performance such as sensitivity mentioned in figure 7(b) is only 14.55%(page 7, line 221) higher. It is unreasonable to conclude that ITIC is suitable for detection.
3. On page 3 lines 98 and 99, “Finally, soda-lime glass was used as a cover material, and it was fixed with a UV-hardened a sealant…”, how can the device be bendable by using rigid soda-lime glass as a cover layer?
4. The data listed in figure 8 are redundant. As figure 8(a) shows, the flexible CsI(Tl) scintillator has a higher luminescence performance. It is natural for CCD and sensitivity to improve when the detector is coupled with the flexible scintillator. This result is related to the product performance of Hamamatsu Company and irrelevant to the experiment and demonstration in this paper.
5. In table 1, power conversion efficiency (PCE) appears as a detection performance parameter. As we all know, PCE is a key parameter to characterize solar cells and is rarely used in detectors. The author needs to give reasons for use and calculation method.

6. At various sections in the discussion, the test conditions are unclear. Conditions such as dark or light stimulates, simulates sunlight or X-rays as excitation source, and bias voltage should be specified in the corresponding section. The test conditions mentioned in the experiment are too confusing to read.
7. The highest sensitivity in this paper is 1.95 mA/Gycm2. What level is this among the reported flexible detectors? A performance comparison table is required. 
8. This paper discusses several major issues including substrates, the type of accptor materials, influence CsI(Tl) scintillator, and the device bending degree. The topic and focus of the discussion are vague. The author should allocate the length reasonably and define the central issue of the essay.

Round 2

Reviewer 3 Report

Dear editor,

The author has responded and revised according to my suggestions. I think the revised manuscript can reach the level of publication.